# Latitudinal Diversity Gradient in the Changing World: Retrospectives and Perspectives

Yu Zhang [1,2], Yi-Gang Song [3], Can-Yu Zhang [4], Tian-Rui Wang [3], Tian-Hao Su [2,5], Pei-Han Huang [1,2], Hong-Hu Meng [1,6,*] and Jie Li [1,7,*]

1. Plant Phylogenetics and Conservation Group, Center for Integrative Conservation, Xishuangbanna Tropical Botanical Garden, Chinese Academy of Sciences, Kunming 650223, China; zhangyu@xtbg.ac.cn (Y.Z.); peihanhuang@foxmail.com (P.-H.H.)
2. University of Chinese Academy of Sciences, Beijing 100049, China; sutianhao@xtbg.ac.cn
3. Eastern China Conservation Centre for Wild Endangered Plant Resources, Shanghai Chenshan Botanical Garden, Shanghai 201602, China; ygsong@cemps.ac.cn (Y.-G.S.); wtianrui@163.com (T.-R.W.)
4. Yunnan Normal University, Kunming 650500, China; zhangcanyu11@outlook.com
5. CAS Key Laboratory of Tropical Forest Ecology, Xishuangbanna Tropical Botanical Garden, Chinese Academy of Sciences, Kunming 650223, China
6. Southeast Asia Biodiversity Research Institute, Chinese Academy of Sciences, Naypyidaw 05282, Myanmar
7. Center of Conservation Biology, Core Botanical Gardens, Chinese Academy of Sciences, Mengla 666303, China
* Correspondence: menghonghu@xtbg.ac.cn (H.-H.M.); jieli@xtbg.ac.cn (J.L.)

**Abstract:** The latitudinal diversity gradient (LDG) is one of the most extensive and important biodiversity patterns on the Earth. Various studies have established that species diversity increases with higher taxa numbers from the polar to the tropics. Studies of multicellular biotas have supported the LDG patterns from land (e.g., plants, animals, forests, wetlands, grasslands, fungi, and so forth) to oceans (e.g., marine organisms from freshwater invertebrates, continental shelve, open ocean, even to the deep sea invertebrates). So far, there are several hypotheses proposed to explore the diversity patterns and mechanisms of LDG, however, there has been no consensus on the underlying causes of LDG over the past few decades. Thus, we reviewed the progress of LDG studies in recent years. Although several explanations for the LDG have been proposed, these hypotheses are only based on species richness, evolution and the ecosystems. In this review, we summarize the effects of evolution and ecology on the LDG patterns to synthesize the formation mechanisms of the general biodiversity distribution patterns. These intertwined factors from ecology and evolution in the LDG are generally due to the wider distribution of tropical areas, which hinders efforts to distinguish their relative contributions. However, the mechanisms of LDG always engaged controversies, especially in such a context that the human activity and climate change has affected the biodiversity. With the development of molecular biology, more genetic/genomic data are available to facilitate the estimation of global biodiversity patterns with regard to climate, latitude, and other factors. Given that human activity and climate change have inevitably impacted on biodiversity loss, biodiversity conservation should focus on the change in LDG pattern. Using large-scale genetic/genomic data to disentangle the diversity mechanisms and patterns of LDG, will provide insights into biodiversity conservation and management measures. Future perspectives of LDG with integrative genetic/genomic, species, evolution, and ecosystem diversity patterns, as well as the mechanisms that apply to biodiversity conservation, are discussed. It is imperative to explore integrated approaches for recognizing the causes of LDG in the context of rapid loss of diversity in a changing world.

**Keywords:** latitudinal diversity gradient; species richness; biodiversity pattern; climate change; conservation; diversification; speciation; extinction

## 1. Introduction

In 1807, Alexander von Humboldt (1769–1859; Figure 1A) proposed the embryonic framework of latitudinal diversity gradient (LDG) and wrote that "The nearer we approach the tropics, the greater the increase in the variety of structure, grace of form, and mixture of colors, as also in perpetual youth and vigor of organic life" [1,2] (Notes: von Humboldt published the first edition of a series of essays that entitled 'Ansichten der Natur' in Berlin, 1807. The essays initiated while he was in South and Central America that was translated variously as 'Aspects of Nature' or 'Views of Nature'. One of the four essays was composed the first edition, 'Ideas for a physiognomy of plants' that contains the following paragraph was translated by Otté and Bohn in 1850). Subsequently, the LDG has been recognized and studied by many biologists, ecologists, and geographers for over 200 years [3–12]. The pattern of LDG is responsible for the broadest and most notable of biodiversity patterns globally (Figure 1B). It has been well documented on the land, in the open ocean, and even discernible in deep sea, which distribution patterns have been characterized for plants, animals, fungi, and marine organisms [13,14]. However, distribution patterns of organisms are not balanced around the globe so that many naturalists and scientists have tried to understand the cause of LDG for centuries [15]. Notably, the LDG is mostly consistent, regardless the geographic context, taxonomic affiliation, or time scale of the biota [16,17]. Previous studies have attested and cataloged many hypotheses to explain the underlying mechanisms that increase species diversity and taxon numbers from high latitudes to the tropics [18]. However, the mechanisms of LDG are not very clear, even with little consensus. It is necessary to review the LDG in the context of a changing world wherein human activity and climate warming are affecting the patterns of biodiversity in evolution and ecology.

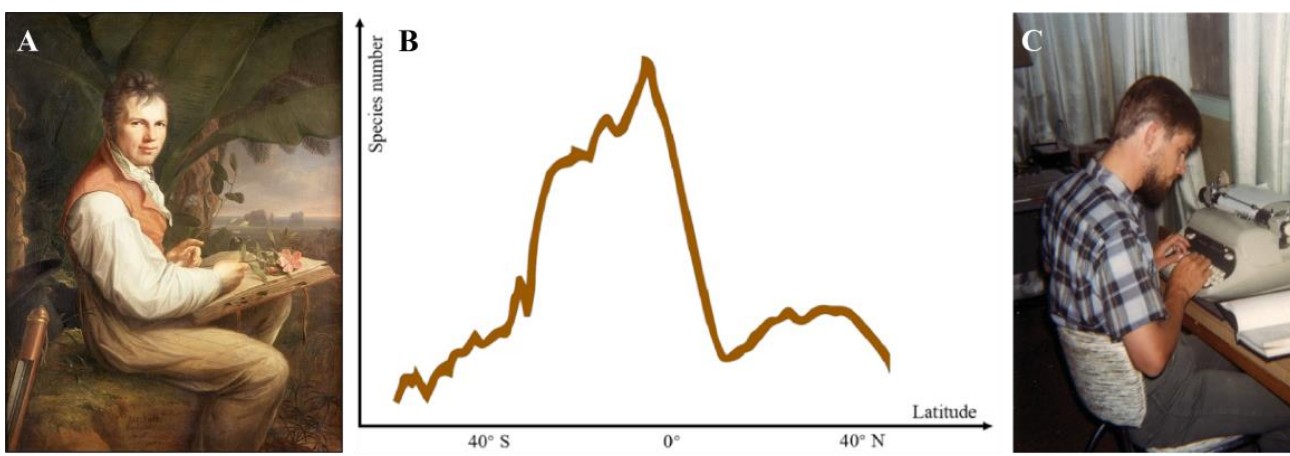

**Figure 1.** (**A**), Alexander von Humboldt (1769–1859) is widely regarded as the "father of phytography". He was the first to state observations about the LDG (photo reproduces from Biogeography [19]); (**B**), diagram of the LDG, with count of species by latitude (diagram adapted from Spatio-temporal climate change contributes to latitudinal diversity gradients [9]); (**C**), Eric Pianka provided insights into the concept and major hypotheses for the LDG in 1966 (Photo courtesy of E. Pianka on 23 February 2022 via Email; this photo has also been used in *Am. Nat.* 2017 [10]).

The investigation of the LDG has drawn on in ecological and evolutionary studies, i.e., some species survive whereas others die out in the process [15]. Almost six decades ago, Eric Pianka (Figure 1C) proposed the first comprehensive review on LDG and the six major hypotheses that compiled a wide range of ideas to explain and address possible causes for patterns in diversity [6,10,15]. Subsequently, LDG explanations have been focused on evolutionary mechanisms [5], such as differences in the time and area available for diversification in tropical and temperate biomes, latitudinal differences in the rates of diversification, speciation and/or extinction in combination with tropical energy, and niche conservatism [10,15]. However, evolutionary biology and ecology must be combined to

explain why larger numbers of taxa are distributed in certain areas of the planet [15]. There is still no consensus on the drivers of LDG that elevate tropical diversity [20].

Alternative LDG hypotheses can be tested with the help of a rich body of biodiversity database, including data concerning phylogeny and biogeography [21]. However, the primary cause of LDG at a global scale is unexplained; and a new synthesis has emerged to determine the geographic ranges of species and their concentration of species within regions, based on evolutionary, biogeographical, and contemporary (i.e., climate and environmental variables) factors [15]. Understanding the underlying mechanisms of LDG is a major goal in conservation biology, biogeography, and ecology [22]. Multiple dimensions of LDG have been described, from intraspecific genetic variation to species richness and phylogenetic diversity, all of which are vital for assessing the underlying processes that shape the distribution of life on Earth and for providing maximum support for global biodiversity conservation [23]. Fascination with the pattern of higher biodiversity in tropical regions has stimulated increasing interest in community ecology [17]. Similarly, the erosion of biodiversity from the local/regional level to the global scale has catalyzed many studies in conservation biology [24]. Thus, the study of the LDG provided a unique opportunity to comprehensively understand latitude-associated patterns in ecology, biogeographical origin, and maintenance of species diversity [25].

Given the potential value of biodiversity conservation efforts, it is critical to examine the mechanisms that create the patterns in biodiversity that produce the LDG. In this review, we aim to identify and discuss the hypotheses for LDG mechanisms, i.e., whether the distribution patterns of different species are consistent with LDG, and explain the factors that cause the number of species to change with latitude. We also discuss the effects and formation of LDG in various periods during which species have been restricted to different latitudes. Additionally, the role of LDG in evolution and ecology will be discussed in relation to global warming and human activity that are affecting the biodiversity. Finally, we concentrate on the relevance of the LDG to biodiversity conservation in the hope that a deeper understanding and reasonable scientific approach (e.g., the big data of genetic/genomic sources) to it can be obtained by studying the LDG.

## 2. Status of LDG from Previous Studies

Pianka's comprehensive review on LDG aided the organization of the myriad hypotheses regarding it in 1966 [6]. Since then, many major journals have published numerous studies about the LDG, such as the flagship journals, *Global Ecology and Biogeography*, and *Journal of Biogeography*, have been the main sources on LDG articles in the past few decades; other mainstream journals including *Ecography*, *Ecology*, *Ecological Letters*, *American Naturalist*, and *Proceedings of the Royal Society B: Biological Sciences*, have also published many papers on LDG (Figure 2). In addition, the number of articles on LDG about land-based flora/fauna, marine organisms, and micro-organisms has increased since 1995, although the trend from 2019 and 2020 has slowed (Figure 3). Overall, the body of LDG-related literatures has grown substantially in recent decades.

All taxa, regardless of the land or the ocean, are conformed to the association between latitude gradients and taxonomic richness, e.g., terrestrial arthropods and terrestrial plants, mangrove trees, birds, mollusks, mammals, corals, freshwater arthropods, marine protists, marine arthropods, and reptiles. The existing distribution patterns of LDG of land-based flora, fauna, micro-organisms, and marine organisms are useful benchmarks to explore the distribution mechanisms so that they are stated briefly as follows.

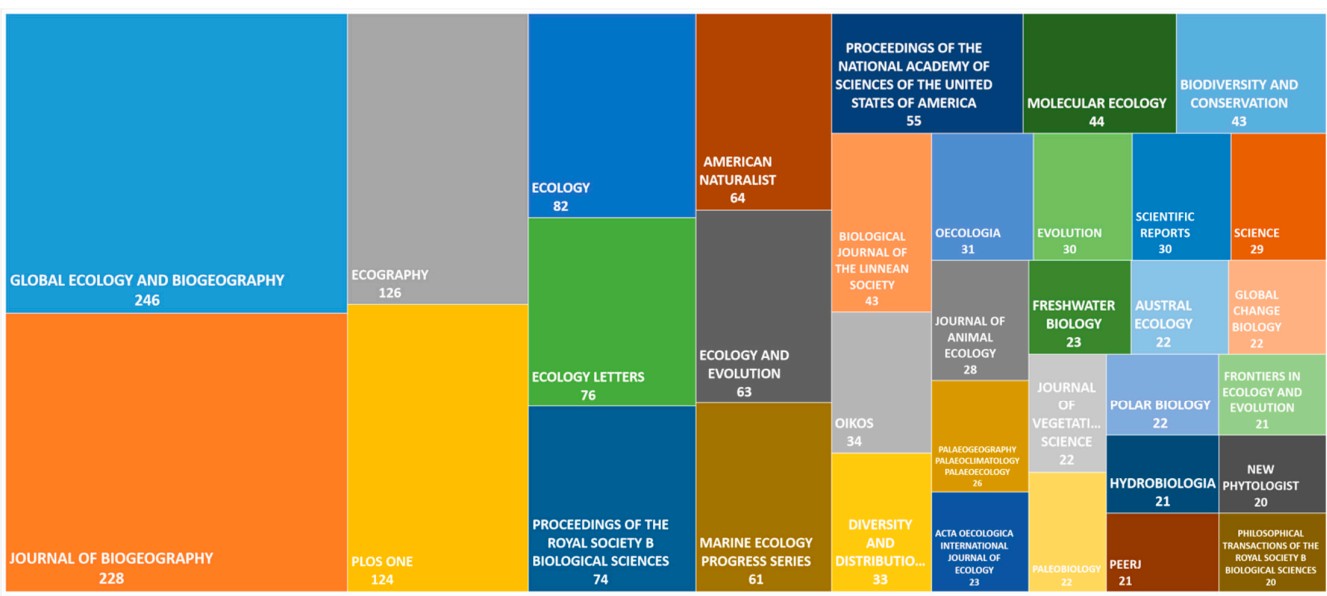

**Figure 2.** Number of documents per journal relating to the LDG research using the keywords "latitudinal diversity gradient" from Web of Science (Data accessed on 22 February 2022); only journals with more than 20 published papers are shown. The size of a block is proportional to the number of relevant papers in the given journal.

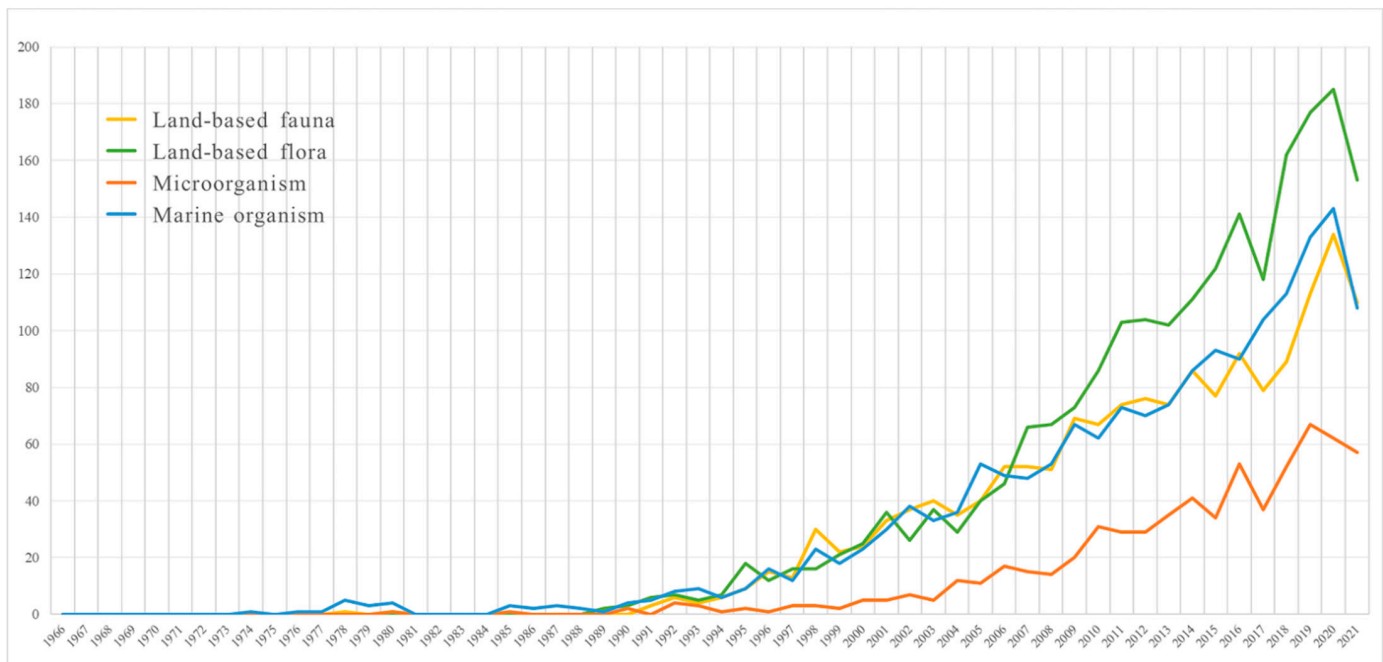

**Figure 3.** Search results using Web of Science (Data accessed on 22 February 2022) from 1966 to 2021, with the numbers of the published papers in each year; results include all documents and trends obtained using the keywords, "latitudinal diversity gradient" and "Land-based flora" (including plant, forest, flora, vegetation, algae, bryophyte, pteridophyte, spermatophyte, or lichen); "Land-based fauna" (including animal, zoology, fauna, vertebrate, invertebrate, bird, fish, mammal, amphibians, reptiles, or insect); "Microorganism" (including microbes, microorganism, bacteria, or germ); "Marine organism" (including marine, ocean, or sea), respectively.

### 2.1. Land-Based Flora

The LDG is most strongly visible in phytogeographical composition. A previous study has observed a similar distribution pattern of all seed plants in China, considering all geographical and climatic variables [26]. Huang et al. revealed that the endemic seed plants show a clear distribution of LDG from north to south, indicating that large-scale phytogeography of endemic flora that is strongly related to latitude, e.g., tropical genera account for approximately 75% of flora species at the southernmost tip (i.e., Hainan), which decreases to nearly 0 at latitude 45–50° N [25]. In Australia, the differences in plant reproductive strategies between communities at high and low latitudes indicate the importance of climate diversity patterns [27]. A global species density map for liverworts showed a significant pattern of LDG in species richness [28]. And latitudinal gradients of the richness of shrub and liana showed the same trends, i.e., the latitudes occupied by shrubs ranged nearly from 18–50° N, a greater range than that occupied by shrubs (18–45° N) and lianas (18–40° N) of latitude in species range size [29]. However, not all plants conform to the obvious LDG, i.e., the latitudinal trend in species richness is weakly negative in some liverworts and woody plants [22]. Notably, climate change is already altering the biogeographical patterns of flora, and the substantially diminish the extent and richness of Europe's high-latitude mountain flora has demonstrated that climate change is predicted to the lost at high latitudes [30]. Therefore, biogeographic histories of flora affected their vulnerability to the climate change, especially the climate warming; and the vulnerability of the endemic plants, implying high significance for conservation decisions in the shifts of LDG.

### 2.2. Land-Based Fauna

The LDG pattern of animals is significant in a broad sweep of taxa, e.g., in the tropics, the squamate lineages originating in situ reflect it, and the patterns are driven primarily by the dispersal and diversification rates [14,31]. Ant species richness also peaks in the tropics, which is consistent with that of many other taxa [32]. Extant birds are globally distributed, although the ubiquity of birds and their penchant for dispersal, and avian diversity studies have shown that both species numbers and the presence of higher taxonomic groups are skewed towards the tropical environments [33]. Approximately 92% of all mammalian diversity peaks in the tropical regions, with the exception of Lagomorpha, which shows maximum diversity in the northern and temperate regions [34]. LDG patterns are strikingly consistent with the diversification rates, wherein the peaks for species richness are always associated with low extinction rates and/or high speciation rates [34].

### 2.3. Microorganisms

The pattern of diversity in microorganisms with a small body size, fast population growth, high abundance, and high dispersal rates, which characteristics are contrary to that of macroorganisms. However, microorganisms unexpectedly showed the LDG pattern existed in bacteria as well as in marine protists (i.e., planktonic foraminifera) on a global scale [16,35,36]. Similarly, the beta diversity and phylogenetic diversity of *Streptomyces* strains showed an LDG pattern [37]. The LDG patterns were speculated to be involved in the geographic distribution of the host organisms. Surprisingly, both host richness and parasite abundance increased across 20° of latitude despite assumptions about diversity in parasites suggesting that their parasites exhibited no pattern or reverse latitudinal gradients to their hosts [38]. The reason for the reverse latitudinal gradients or lack of pattern in LDG in microorganisms is that the greater areas of wetlands at higher latitudes provide many habitats for larval amphibians to enhance host density, contributing positively to parasite richness [39]. In different organism groups (e.g., meiofauna, zooplankton and unicellular taxa), there is a significant difference in the LDG pattern between species richness and latitude [35].

### 2.4. Marine Organisms

The existence of LDG patterns in the sea is surprisingly controversial, especially when land organisms show pervasive LDG with maximum species richness in the tropical regions [40]. However, the LDG patterns for known marine organisms are well-studied in many taxa. Although the extant data from the deep seas are insufficient to analyze the LDG for sparsely distributed deep-sea organisms, some marine epifauna resident on the surface of the substratum in the ocean show a typical LDG [40]. The diversity of nematodes shows a positive LDG in the deep sea in the Atlantic, with a decline from 0° to 40° S [41]. The eastern Pacific bivalvia show a strong LDG within increasing numbers of species from the tropics to the southern tropical boundary in the northern Hemisphere; species numbers outside of the tropics show a stepwise decline toward the poles from 5° S to 8–9° N [40]. Interestingly, the fossil records of marine bivalves from the three successive slices in the late Cenozoic showed that species with tropical origins tended to expand from the tropics to higher latitudes [13]. These results support LDG in the marine shelf benthos, which is congruent with the diversity trends of gastropods along both northeastern Pacific shelves and the northwestern Atlantic [13,42]. For the coastal plants, e.g., mangrove, the mixing-isolation-mixing cycles can potentially generate species at an exponential rate, thus combining speciation and biodiversity in a unified framework by permitting intermittent gene flow during speciation [43]. But, the LDG pattern is still unclear for this.

### 2.5. LDG and Biodiversity Conservation

Biodiversity enhances many natural resources that are essential for human well-being, yet human activity has resulted in rapid biodiversity loss [44]. The sixth mass extinction in Earth's history has been driven by human activity [45,46] and global warming [44,47,48]. Accelerated biodiversity loss is a hallmark of the Anthropocene, in which large declines in population size, habitat loss, fragmentation, biological invasions, pollution, and climate change have been widely observed requiring effective and efficient conservation managements [45,48–51]. LDG, as a wide-scale diversity pattern on Earth, has inevitably been affected by the changing environment. The spatial model indicates that forests and jungles are exposed to anthropogenic threats (e.g., changes in fire regimes and deforestation) in Amazonia, Central America, the Eastern Arc Mountains, the Northern Andes, the Brazilian Atlantic, southeastern Asia, and Sub-Saharan Africa [23,52]. The pattern of LDG will inevitably be influenced in this dynamic environment; hence, the changing trends of biodiversity loss to human activity and climate change in the Anthropocene require more attention.

However, our understanding of the full impact of humanity on biodiversity, as well as of the links between the processes occurring in natural ecosystems, is incomplete [51]. As an important indicator of biodiversity patterns, the LDG patterns will be affected by climate warming and human activity over the global biodiversity framework. Over the past few decades, scientific studies have played important roles in verifying and identifying explicit goals for plant conservation, which are used in assessments of extinction risk, the prediction of range changes under climate change, and adaptation measures [30,53–56]. Therefore, we suggest that the cohesive nature of species richness and ecosystem diversity, particularly the trends of LDG in the changing world, will provide important insights into prioritizing conservation efforts.

## 3. Formation Mechanisms of the LDG

### 3.1. LDG Hypotheses

For the formation mechanism of LDG, there are six main hypotheses in a comprehensive review of LDG have been proposed previously, and these mainly focus on ecology and evolution [6,10]. The hypotheses from 1966 to 2021 for the latitudinal diversity gradient and the other sources are listed in Table 1. With the increase in the knowledge about LDG, more hypotheses have been proposed, but the late-comers are mainly deduced from the six hypotheses. In this review, we simply elaborate the relationships between the six main

hypotheses and the others; and mainly focus on the widely embraced hypotheses. Thus, in this review the six broadly accepted hypotheses are revisited.

**Table 1.** Hypotheses from 1966 to 2021 for the latitudinal diversity gradient, P1–P6 from Pianka (1966), F1–F5 from Fine (2015), and O1–O7 from the other sources.

| Hypothesis | Primary Focus | References |
|---|---|---|
| P1. The time theory | Ecology and evolution | [6] |
| P2. The theory of spatial heterogeneity | Ecology | [6] |
| P3. The competition hypothesis | Ecology | [6] |
| P4. The predation hypothesis | Ecology | [6] |
| P5. The theory of climatic stability | Ecology and evolution | [6] |
| P6. The productivity hypothesis | Ecology | [6] |
| F1. Time-integrated area, energy, and tropical niche conservatism | Evolution | [15] |
| F2. Climate stability | Evolution | [15] |
| F3. Temperature and evolutionary speed | Evolution | [15] |
| F4. Biotic interactions and speciation rate | Evolution | [15] |
| F5. Biotic interactions and finer niches | Ecology | [15] |
| O1.The ecological regulation hypothesis | Ecology | [32] |
| O2.The "diversification rate hypothesis | Evolution | [57] |
| O3.The out of the tropics hypothesis | Ecology and evolution | [58] |
| O4.The out-of-the-extratropics hypothesis | Ecology and evolution | [59] |
| O5.The evolutionary time hypothesis | Evolution | [32] |
| O6.The time-for-speciation hypothesis | Evolution | [60] |
| O7. The tropical niche conservatism hypothesis | Ecology | [61] |

As known, the other hypotheses originate from the six main hypotheses and make differing predictions about the spatial distribution of organisms. For example, the out of the tropics (OTT) hypothesis describes how the combination of evolutionary dynamics and dispersal may have shaped the LDG of marine species [58]. Later, some groups may have also been shaped by dispersal towards the tropics, as in the out of the extratropics (OET) hypothesis [59]. Similarly, the evolutionary time hypothesis (ETH) suggests that tropical areas have been occupied for longer than temperate region, and thus have had more time to accumulate species [32]. Alternatively, under the diversification-rate hypothesis (DRH), higher richness in some clades is explained by faster rates of net diversification, and high species richness is associated with clades that have accumulated many species in a relatively short period of time [57]. The ecological regulation hypothesis (ERH) posits that there are equilibrated the ecological limits to species numbers, which vary systematically with latitude, perhaps due to the direct influence of climate and/or available energy [32]. The tropical conservatism hypothesis (TCH) suggests that the tropics have been occupied for longer, dispersal out of the tropics is rare, and the greater past area of the tropics yielded more present-day tropical clades [61]. From the above-mentioned examples of hypotheses, each of them is related with one or two of the six main hypotheses. Thus, the review briefly elaborates on the possible causes and/or hypothesis of LDG as follows.

The time hypothesis is perhaps the most widely accepted and the oldest of the six hypotheses. It can be dated back to the time of Alfred Russel Wallace, who proposed that evolutionary (speciation) and ecological (immigration) factors drove the increase in species richness of communities over time [6]. In tropical regions where have been occupied for longer periods, yielding more present-day tropical clades, and dispersal out of the tropics is rare [62]. The evolutionary time hypothesis suggests that the tropics provide more time for lineages to accumulate species [21]. This can be explained by geological events, i.e., the tropical regions remained relatively undisturbed, whereas the diversity in northern latitudes was reduced due to multiple glaciations, which led to the formation of the current LDG patterns [10].

The environmental heterogeneity hypothesis suggests that habitat heterogeneity promotes species richness [63], and the probability of species coexistence augments different niches [64]. Heterogeneity is beneficial in case of adverse environmental conditions. For the environmental heterogeneity, it has been shown that communities with increasing species capability and speciation withstood isolation or adaptation to diverse environmental conditions [65,66]. In addition, lower species ecological tolerance in the tropics may result in denser speciation and spatial heterogeneity at lower latitudes [54].

The competition hypothesis was proposed based on natural selection in the temperate zone, which was controlled by abiotic more than biotic factors with stronger biotic interactions and increased speciation rates for positive feedback; niches are narrower, competition is stronger, and co-evolutionary rates across the geographic mosaic have reduced the extinction rates [6,10,15]. Therefore, competition in the tropical regions is lower, as intense predation in the tropical environments results in reduced populations.

As an alternative to the competition hypothesis, the predation hypothesis suggests that competition in the tropics is lower due to higher predation in tropical environments, which causes a reduction in population size [6,10]. From the polar to tropical regions, more diversification was observed in the community composition and structure that are the greater probability that a larger proportion of predators can be maintained. Predators can then effectively control the number of prey and producer populations, such that there are more predators, parasites, and prey in the tropics than in other regions [67]. In tropical forests, trees attract their consumers, so that the consumption of seeds and seedlings by animals/herbivores reduces the number/survival rate of seeds/seedlings, and consequently the density of the population. In this way, herbivores increase the space available for the invasion of seeds and seedlings of other plant species, concomitantly increasing the diversity of tree species in the tropical forests [68].

The climatic stability hypothesis predicts that the tropical regions have a higher species richness, greater specialization, and narrower niches due to their stable climate [6,9,10]. The stable climate in tropical regions increases species richness; for example, climate oscillations have affected species diversity globally especially during the Late Quaternary period [69]. However, in the tropical regions with relatively stable climate trends that are likely to have prevented large demographic fluctuations, thus promoting the maintenance of species richness and intraspecific genetic diversity [69,70]. By contrast, rapid climate change in temperate and cold and/or arid regions has led to more profound effects on precipitation and temperature trends, which has directly or indirectly affected species demographics over time [69,71–73]. LDG patterns of increasing latitudinal ranges in animal and plant species richness from low to high are related to the tolerance to seasonal temperature variability [11,74] and Ice Age temperature fluctuations [75].

The productivity hypothesis suggests that species richness increases because of the greater productivity in tropical regions, allowing tighter species packing and narrower niches with a greater overlap of niches [6,10]. Energy input may enhance mutation and physiological rates, which then increase speciation by decreasing generation times [76]; thus population sizes should also correlate positively with productivity [77]. The net primary productivity and constraint on species richness due to limited resources may lead to geographical variation in species diversity [7,78]. Plant richness is primarily limited by water availability and solar energy at the base of the global food web [77,78]. For example, higher amounts of energy lead to faster evolutionary rates and more species richness in flowering plants [54]. In turn, predator richness is limited by the secondary production of herbivores in the food chain, whereas herbivore richness is limited by the net primary production of plants [78]. A positive relationship between richness and productivity could be responsible for the LDG pattern; however, the productivity hypothesis has not been accepted as an important cause of the LDG [17].

### 3.2. Climate Change, Temperature, and Precipitation

There is an equilibrium ecological limit to species numbers that varies systematically with latitude in LDG patterns due to the direct influence of available energy and/or climate [79,80]. The LDG mechanisms are difficult to determine due to the strong correlations between related ecological parameters, including climate, latitude, and temperature [37]. The species richness often varies with elevation and latitude or between geographic regions and temperature, precipitation, or other factors [7]. That is, the effects of species distribution on different species reflect different climatic tolerance among them. Thus, the explanations of the climatic variability hypothesis for variation along the latitudinal gradient favor the evolution of broader climatic tolerance of species at high latitudes [29,30].

During the geological ages, i.e., between the Tertiary and early Eocene period increased, global atmospheric temperatures increased and taxa evolved in tropical regions (low latitudes), and dispersed to higher latitudes, showing latitudinal patterns through tropical areas to higher latitudes [26]. And the current day species richness is also affected by climate oscillations on species demographic during the Late Quaternary [69,81]. Trends have been observed in the changes in the relative frequencies of tropical and temperate genera along the temperature, precipitation, and radiation gradients along the LDG [25]. For most plants, climate cooling (i.e., freezing tolerance) created an evolutionary barrier or survival limit, which determined the distribution range of many flowering plant taxa. For example, cold-intolerant plants of the boreotropical and evergreen flora were forced southward by climate cooling in the northern hemisphere [82]. Thus, this evolutionary process for cold adaptation is reasonable, and the tropical genera in flora descend along latitudes from low to high [26]; even some genera colonized from warm to cold [83]. An understanding of the environmental aspects that influence the traits underpinning adaptive resilience to changing climates could help in assessing the vulnerability of populations to climate change [84,85].

Intraspecific genetic diversity and high levels of species richness are also correlated with the past inter-annual precipitation variability [23]. For example, frequent variations in precipitation during the Late Quaternary and the resultant fluctuations are proposed to have driven population isolation and adaptive divergence in suitable habitats at lower latitudes [86,87]. Precipitation can be used to explain the patterns of selection of local and regional variations in climate regimes [88].

Model selection and hierarchical partitioning of species richness in liverworts showed that water-related variables are dominant [89]. For precipitation seasonality and availability, the spatial precipitation heterogeneity in mosses is considered an important predictor of species richness, but the temperature variables generally have higher explanatory power than water variables in woody plants [22]. Also, hierarchical partitioning in the species richness of liverworts and mosses indicated that the independent effects of temperature variables are higher than those of water variables and that water variables have more variation in these families than in woody plants [29]. This is consistent with the evolution of terrestrial plants, which involves their ability to adapt to water deficiencies [89]. Additionally, population persistence due to long-term climate stability results in a higher accumulation of species richness in the tropics than at higher latitudes, and climate-driven processes at lower latitudes always result in higher population divergence due to frequent precipitation variability [23].

### 4. Evolutionary Responses for LDG

Generally, evolutionary responses are prerequisites for the long-term persistence of biodiversity during ongoing and projected scenarios of the extreme climate events [85,90]. Many LDG results from ecological, historical geology, and demographic events that influence dispersal and diversification can be explained by the evolutionary responses based on the historical contingency proposed [62,91]. Likewise, the diversification rate suggested that extinction and/or speciation rates vary systematically with LDG [32]. Evolutionary responses (i.e., speciation, extinction, diversification, and dispersal rates) are inevitable in

LDG patterns. The evolutionary forces generated by LDG have been debated for many years [5,62]. Here, related evolutionary responses are listed to address the LDG pattern.

### 4.1. Speciation Rate

The key to explaining LDG patterns is to understand the variation in speciation and extinction rates with latitude; however, it is difficult to estimate the diversification rates associated with specific geographic locations [34]. The speciation rates differ due to a range of factors and in different geographic regions [7]. In tropical clades, speciation is relatively high, and extinction appears to be very low, whereas in temperate lineages, speciation and extinction are very high [31,92]; the speciation rate is always linked to the diversification rate. For example, the relative stability in tropical climes led to older, slower-evolving but still species-rich communities, and long-term cooling had a disproportionate effect on the non-tropical diversification rates, leading to dynamic young communities outside of the tropics [93]. Thus, nonequilibrium explanations for LDG have been proposed based on different speciation rates in the tropics [35]. For example, in tropical regions where the ambient conditions are warmer, environmental factors could increase the mutation rates and lead to faster rates of evolution [8,94]. Thus, the speciation rate affects LDG patterns.

### 4.2. Extinction Rate

The extinction rates have led to geographic patterns of species richness; for example, a geographic region with climatic stability may play a key role in extinction levels [7]. In addition, species with larger distributions are less prone to extinction, and regions experiencing intense climatic fluctuations always experience increased extinction rates [70,95,96]. In the phylogeny, the older crown groups from the phylogenetic tree always accumulate more species as expected, whereas the stem groups have fewer species so that older temperate taxa have an apparently attenuated extinction [31,92]. And in the tropics, the mammals, e.g., amphibians, and squamates, also have lower extinction rates that contributed to more net diversification than speciation [31,34,97]. The dependence of extinction and speciation on diversity is a general process that regulates the shapes of taxonomic diversity curves [5]. Together, the association between a high species richness and low extinction rate suggested that extinction may play a more important role in driving differences in net diversification rates than speciation along latitudinal gradients [7].

### 4.3. Net Diversification Rate

The outcome of both speciation and extinction is the net diversification rate, which is typically higher in tropical clades and represents the difference between speciation and extinction [31]. The rates of diversification are higher in tropical latitudes; a strong, consistent role of net diversification in driving latitudinal species richness gradients has been proposed [7,8,71]. Net diversification rates suggest that higher temperatures or energy fluxes promote more rapid evolution; and a more equitable climate favors habitat specialization, leading to the dispersal and gene flow reduced and/or more intense biological interactions drive the adaptive changes [6,11,54,98]. Notably, the areas with high net diversification rates are more likely to be evolutionary cradles [14]. In the tropics, the squamate lineages suggested the greater species richness due to speciation, but the LDG appears to be driven primarily by the diversification rates [31]. Nevertheless, there are exceptions; e.g., the clades of passerines and swallowtails showed a significant latitudinal effect on the relative diversification rates [99].

### 4.4. Dispersal Rate

Dispersal always promotes gene flow wherein the colonization of habitats among isolated patches occurs. It has played a key role in species and population persistence in fragmented systems under rapid climatic change [81]. Two main hypotheses on dispersal dominance have been proposed. Firstly, the "out of the tropics" hypothesis suggests that lineages originate and diversify in the tropics and disperse from the tropics to the temperate

regions. Secondly, the "tropical niche conservatism" hypothesis suggests that the lineages originate in the tropics and accumulate in tropical regions, because it is difficult to disperse and adapt in temperate regions [13,34]. Most of the present-day diversity in marine bivalves in extratropical regions is from taxa shared with the tropics, which has supported dispersal and the persistence of an LDG [40,58]. The role of dispersal in generating diversity gradients has been determined by studying the frequency of shifts from temperate to tropical biomes and vice versa [9]. More lineages are dispersed from the tropical to temperate regions than from the temperate to tropical regions, and the dominance of tropical to temperate dispersal is statistically significant in all scenarios [9,12,13]. In tropical regions, the newly evolved taxa might have extended northward into the temperate regions [100]. However, the range of extension in warm regions was curbed by frost tolerance; rapid and large-scale range contractions and expansions may have resulted in population extirpation and the subsequent loss of species richness at these latitudes [80,81].

## 5. Future Perspectives

The most unique feature of the Earth is the existence of life, and the most extraordinary feature of life is its diversity (i.e., genes, species, and ecosystem diversity), which provides numerous essential services to a human-dominated society [48,49]. Considering that LDG is the broadest and most notable biodiversity pattern [15], the conservation efforts should be linked to the mechanisms of ecology and evolution from genetic/genomic diversity, species diversity, and ecosystem diversity in the future projects.

In recent decades, the studies of LDG have evolved due to the promotion of studies on land-based flora/fauna, microorganisms, and marine organisms (Figure 3). Using the comprehensive review written by Pianka as a milestone [6,10], the myriad of hypotheses for LDG has been organized into a manageable framework for future studies. Many studies on LDG have developed scientific tools for the elucidation of diversity and the related causes. However, most biodiversity patterns of LDG have mainly been explored based on ecosystem and species numbers [10,15]. As known, species is the evolutionary unit, and species diversity is the basic element for evolutionary change. Also, the ecosystem diversity of specific areas preserves many species and the subsequent genetic diversity [48]. According to the Convention on Biological Diversity (CBD, www.cbd.int, accessed on 23 January 2022), genetic diversity is recognized as one of the three basic elements of biodiversity and is the focus of many conservation genetics studies. Thus, we suggest that LDG patterns are useful for investigating the ecological and evolutionary mechanisms from genetic/genomic and phylogenetic diversity to facilitate the estimation of clade-specific diversification rates in relation to latitude, climate, and other factors. Estimating the contribution of these factors from ecology and evolution in promoting the LDG patterns from phylogenetic and/or genetic/genomic data is challenging, but the availability of big data is gradually reality. Also, it is worth noting that the development of Geographic Information System (GIS) mapping and satellite imagery has provided unprecedented resources to study LDG from the perspectives of global patterns of climate, productivity, landform, and species richness [10]. In addition, LDG can be explored with technological achievements from both paleo- and modern studies [18].

With the coming of new era of the Anthropocene, the biodiversity crisis is closely connected to the human activity. Predicting the influence of human-induced climatic change on a short and/or long-term organismal distribution is imperative in contemporary biology [101]. Over the course of a century, humans have markedly altered the planet, causing various effects, including an increase in ocean acidity to landscape fragmentation and climate change [48]. The far-reaching influence of human activity has contributed to a loss of biodiversity, changing land use, habitat loss, plant extinction, predatory fish, defaunation, and a reduction in species abundance [48,102]. Therefore, investigations of the LDG pattern must consider the knock-on effects on biodiverse communities from the concomitant influence of human activity and climate change. According to the shifts or trends of LDG patterns, some protected areas can be identified. Protected areas are

safeguarded from human activity to a certain extent through conservation planning and prioritization in a human-dominated and fast-changing world [48]. Now the solution of human activity to biodiversity is easily found to these issues, i.e., the establishments of related legislations and conservation areas to decrease the impacts from human activity. However, little is known about the effects of climate change on biodiversity, and thus, significant research on this aspect is crucial. Previous research on well-studied large organisms showed that biodiversity may be more sensitive to climate, such that the impact of ongoing Anthropocene climatic change may be much more serious than previously thought [58]. In particular, wide-scale extinctions and population decline across taxonomic groups have been caused by human activity over the past 500 years [103]. However, most studies on LDG have mainly focused on ecological factors, i.e., species and ecosystems. Therefore, the impact of human activity or imprint and climate change (e.g., the warming climate) on the diversity of ecosystems along latitudinal gradient has been generally disregarded.

In addition, the explanations of LDG patterns and/or mechanisms from genetics and genomics are rare. The reasonable utilization and benefit of sharing of genetic resources have been inferred from the CBD to ensure the conservation of biodiversity [104]. Based on whole genomes, a strategy for cataloging adaptive genetic diversity to climate change across a range of ecologically important non-model species has improved population datasets and provided a high-resolution record of variation in structural information and genomes [88]. The models linking genomics with eco-evolution provide unique opportunities for predicting and tracking vulnerability and adaptive responses to climate change, which will benefit the biodiversity issues along the LDG distribution. Thus, LDG taken from large-scale genetic/genomic data (e.g., the DNA data from NCBI) or the genetic variation in specific taxa in wild populations with a vast distribution will reveal the distribution patterns and mechanisms to assist the CBD in meeting targets to halt the acceleration in biodiversity loss. In the face of environmental change, the ever-increasing availability of ecological studies is going on, integrating the big data from evolution across large-scale distributions and taxa, will provide important and novel opportunities to enhance our understanding of the adaptive potential of LDG globally. Hence, ecologists, evolutionists, and related scholars are called upon to rethink the LDG in view of the available genetic resources in the changing world.

## 6. Conclusions

Half a century ago, the review from Pianka is believed to be a milestone in understanding LDG patterns, which is and will be the manageable framework for future studies [6,10]. Subsequently, a time-integrated biogeographic analysis of geographical diversification suggests that both time-integrated and stable climate areas will determine the baseline of marine diversity and terrestrial patterns at the global scale. Patterns of the LDG mainly focused on the ecology and evolution in tropics from the species richness to mechanism. The tropics are geologically older, and have had more time for diversification, which is consistent with the time hypothesis, in which biotic interactions likely augment coexistence and diversification [15]. Thus, the related studies from tropical regions to warm and/or cold regions are the key points we have to take into account, e.g., the different taxa and/or communities along with latitudinal gradients. According to our own experience, *Engelhardia* is distributed from 10° S to 30° N on a latitudinal distribution in Southeast Asia, where the plants are typical to unique substrates to disentangle the LDG from integrated disciplines [100,105].

Here, this review presents a systematic overview of the broad and comprehensive literatures on the state of LDG in a changing world where ecology and evolution have been applied to its distribution patterns. In the future, we suggest that genetic/genomic-based approaches on LDG should be integrated for the understanding of biodiversity conservation. This facet is still underdeveloped, and the numbers of studies that elaborate biodiversity patterns of LDG based on the genetic/genomic data are still scarce, especially regarding the relationship between the variations in genetic/genomic data and the

pre-existing data from ecosystems and speciation (e.g., phylogeny and genetic diversity). Moreover, the use of molecular studies (i.e., genetic/genomic diversity) is important to highlight the potential for the establishment of theories on LDG that do not have sufficient support from genetic/genomic implementation, because the dataset for a considerably large-scale distribution is difficult to obtain, especially the vast regions across oceans. However, genetic/genomic criteria of specific taxa with a vast distribution are useful and possible in some plants with a large latitudinal distribution if the field works is comprehensive. The large-scale taxa collection combined the big data DNA information, will enable the CBD to ensure biodiversity conservation through the sharing and utilization of benefits from genetic/genomic resources. Thus, exploring the LDG from large-scale DNA data to disentangle the diversity mechanisms and patterns to inform biodiversity conservation and management measures may be a valid approach. Comprehensive biodiversity patterns, together with the determination of ecological and evolutionary mechanisms should therefore be used to understand the mechanisms and causes of LDG in a changing environment where biodiversity is rapidly declining and disappearing.

**Author Contributions:** Conceptualization, Y.Z. and H.-H.M.; Funding acquisition, H.-H.M. and Y.-G.S.; Project administration, H.-H.M.; Visualization, Y.Z. and H.-H.M.; Formal Analysis, Y.Z.; Writing—original draft, Y.Z., Y.-G.S., C.-Y.Z., T.-R.W., T.-H.S., P.-H.H., J.L. and H.-H.M.; Writing—review and editing, Y.Z., Y.-G.S., C.-Y.Z., T.-R.W., T.-H.S., P.-H.H., J.L. and H.-H.M. All authors have read and agreed to the published version of the manuscript.

**Funding:** This research was funded by National Natural Science Foundation of China (No. 42171063); Southeast Asia Biodiversity Research Institute, Chinese Academy of Sciences (No. Y4ZK111B01); the Special Fund for Scientific Research of Shanghai Landscaping & City Appearance Administrative Bureau (Nos. G192422 and G212406); Youth Innovation Promotion Association, Chinese Academy of Sciences (No. 2018432); and the CAS "Light of West China" Program.

**Institutional Review Board Statement:** Not applicable.

**Data Availability Statement:** Not applicable.

**Acknowledgments:** The authors are grateful to Chong-Rui Ai (Xishuangbanna Tropical Botanical Garden, Chinese Academy of Sciences) and Hui-Jie Qiao (Institute of Zoology, Chinese Academy of Sciences) for their kind assistances and suggestions in the references & literatures investigation for this paper.

**Conflicts of Interest:** The authors declare no conflict of interest.

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
