# Peer review of "Latitudinal Diversity Gradient in the Changing World: Retrospectives and Perspectives"

_diversity, doi:10.3390/d14050334_

Round 1

Reviewer 1 Report

In this paper, Zhang et al. comprehensively summarized the insights of Latitudinal Diversity Gradient (LDG) from the Pianka’s comprehensive review to the recent progress. It has been well acknowledged that the LDG is an important issue and its patterns are very common in different organisms. The related studied of LDG are too many, however, the mechanisms of the LDG are not clear though there are increasing hypotheses or explains for such vast patterns in biodiversity globally. Pianka’s article in 1966, was a milestone in organizing the myriad hypotheses for the LDG into a manageable framework for future study. What will the next decades, even the next 50 years bring? We need a look forward against the background that biodiversity crisis, but the related studies will benefit from the availability of big data. Thus, this review is useful to understand the diversity at large scale and suitable to Diversity so that I recommend the minor revise.

The questions I concern are:

  1. The English of the manuscript need to be revised, because the current form still have the errors, and mistakes;

  1. In 1807, Alexander von Humboldt (1769–1859) proposed the theory of LDG, I think the related expression of Humboldt, “The nearer we approach the tropics, the greater the increase in the variety of structure, grace of form, and mixture of colors, as also in perpetual youth and vigor of organic life”, is the prototype of LDG, not the theory. Similarly, the expression in the manuscript should be rigorous in a scientific paper, so the authors need to revise the whole manuscript, making all terms to be more accurate.

  1. Figure 1 should be redrawn to bring them into harmony, e.g., A and C are colorful, B is black-white, so the authors should change the plate B into colorful.

  1. After the retrospectives, and the perspectives in the 4th section, I suggest the authors to synthesize a more specific but more applicable in ecology, evolution and biodiversity conservation. This is very important in a review article, and the well-written synthesis will attract broad audiences.

  1. In tropical clades, speciation is relatively high and extinction appears to be very low, whereas in temperate lineages, speciation and extinction are very high. Actually, this is not always the case, see Sun et al., 2021, Nature Communications, https://www.nature.com/articles/s41467-020-17116-5.

Author Response

Dear reviewer,

All your concerns about our manuscript have been respoded point-by-point, please check the attached word.

Best,

All author

Reviewer 2 Report

LDG is the broadest and most notable of biodiversity patterns,naturalists and scientists have tried to understand the cause of LDG for centuries. Human activity and climate change caused the loss of biodiversity, but how to affect LDG is unknown. Although have some reviews about LDG, they mainly focus on specific groups, and there is no evaluation of the whole biodiversity. This manuscript makes a good summary in time. The manuscript has a clear train of thought and the literature is comprehensive, which is a good summary, but some specific details need to be further revised.

Sincerely

Author Response

(The authors gave the same response as above.)

Reviewer 3 Report

After the reading of the whole text, my concerns of the manuscript are listed as follows:

Line 2, (LDG) can be deleted;
Line 20-21, this sentence should be rephrased;
Line 26, this sentence “We reviewed the progress of LDG”is not completed;
Line 28, Evolution, should it be referred in the LDG?
Line 33-36, this sentence is not clearly, should be rephrased;
Line 37-38, pls consider the evolutionary mechanism, and this had been included in the main context;
Line 48, the LDG pattern?
Line 51, The phenomenon …..? maybe pattern is better.
Line 53, The LDG? And “the” should be added before LDG, because here should be a specific phrase or noum, pls check the related in the whole context;
Line 61-62, this sentence should be cleared;
Line 70, ecology or, another expression here? May be mainly on ecological study or ecology;
Line 84, delete “now”;
Line 101, which? 
Line 117, Global Ecology, Biogeography? Wrong here!!!
Line 216, this sentence should be rephrased;
Line 227, Therefore?
Line 322, in the section, the authors gave main six hypothese, and explained the reason why the other hypotheses did not include in the main text. However, I suggest here need more specific reason in this section;

Line 455, in this section, “Future perspectives”, the related synthesis should be more concise, and some viewpoints should be directly pointed here.

Author Response

(The authors gave the same response as above.)

Reviewer 4 Report

Revisited Latitudinal Diversity Gradient (LDG) in the Changing World: Retrospectives and Perspectives

Yu Zhang1,5, Yi-Gang Song2, Can-Yu Zhang3, Tian-Rui Wang2, Tian-Hao Su4,5, Pei-Han Huang 1,5, Hong-Hu Meng1,6,*, Jie Li1,7,*

Review of the manuscript

The review contains 98 literature sources. It investigates the topic from a recent perspective focusing on possible changes due to the changing world.

The review is valuable and adequate for the profile of the journal, therefore it is suggested for publication.

A small change is suggested in the title to the following:

„Latitudinal Diversity Gradient (LDG) Revisited in the…”

The indication of the aim is at the end of the introduction (lines 101-110). Some rewording is necessary for replacing phrases expressing „what was done” to „what is/was going to do”, e.g. change „In this review, we identify” to „In this review, we aim to identify” (line 102) or „Furthermore, we discuss” to „Furthermore, we are going to discuss” (line 105), etc.

The references should be listed in the sequence of occurrence in the text and indicated by a continuous numbering in the references and in square brackets in the text.

Figure 2 is informative but the letter size should be increased, especially towards the smaller blocks it is impossible to read.

A few small mistakes

line 68

„photo also been” – change to „photo has also been”

line 115

„are shown.” – change to „are shown. The size of a block is proportional to the number of papers in the given journal.

[if it is so indeed]

line 175

Streptomyces – should be in italics

line 235

„hypotheses are from” – change to „hypotheses are originating from”

line 249

„due tothe ” – change to „due to the” (missing space)

line 254

„hypothysis” – change to „hypothesis”

line 290

„...structure, the” – change to „....structure are, the”

lines 293-294

„In tropical forests, trees attract predators such that animal predation of seeds and seedlings is reduced to ……………” – change to „In tropical forests, trees attract their consumers, so that consumption of seeds and seedlings by animals/herbivores is reducing the number/survival rate of seeds/seedlings, consequently the density of the same kind of trees, and in this way provides space for the invasion of seeds and seedlings of other plant species, concomitantly increasing the diversity of tree species in the tropical forests [xx].”

This long sentence could be formed into 2 sentences:

„In tropical forests, trees attract their consumers, so that consumption of seeds and seedlings by animals/herbivores is reducing the number/survival rate of seeds/seedlings, consequently the density of the same kind of trees. In this way herbivores increase the space available for the invasion of seeds and seedlings of other plant species, concomitantly increasing the diversity of tree species in the tropical forests [xx].”

line 489

„landscape use” – change to „land use

line 500

„extirpations” – change to „extinctions”

Author Response

Dear reviewer,

All your concerns have been responded point-by-piont, please check the attached word.

Best,

All authors

Reviewer 5 Report

Dear Authors,

I read with interest your manuscript. Indeed, putting some order in the hypotheses on LDG is an interesting effort, and in my opinion deserves to be published in Diversity.

I have only a few remarks. The first one is about the introduction. It seems to me that it is a little bit confusing, and the same concepts are popping out in different portions of it. Maybe it could be rearranged in a more fluent way. A second remark is related to the second section of the manuscript, where status of LDG from previous studies is analyzed. As far as I know, in plant sciences (I am a botanist), research on the LDG topic is particularly active, while section 2.1 cites 6 manuscripts only. Simply querying google scholar for the last 3 years returns far more results. I would have appreciated a wider review of literature in this part of the manuscript.

I also suggest you to have your manuscript reviewed by a native English speaker, since I noticed some minor issues which could be addressed to make reading the manuscript more pleasant. Plus, there are some errors, such as (but not only):

Line 116: maybe “Since” should be removed

Line 249 tothe -> to the

Best regards

SM

Author Response

(The authors gave the same response as above.)
